# Microbial Acoustical Analyzer for Antibiotic Indication

**DOI:** 10.3390/s22082937

**Published:** 2022-04-12

**Authors:** Boris Zaitsev, Irina Borodina, Ali Alsowaidi, Olga Karavaeva, Andrey Teplykh, Olga Guliy

**Affiliations:** 1Kotelnikov Institute of Radio Engineering and Electronics, Russian Academy of Sciences, Saratov Branch, 410019 Saratov, Russia; borodinaia@yandex.ru (I.B.); teplykhaa@mail.ru (A.T.); 2Institute of Biochemistry and Physiology of Plants and Microorganisms—Subdivision of the Federal State Budgetary Research Institution Saratov Federal Scientific Centre of the Russian Academy of Sciences (IBPPM RAS), 410049 Saratov, Russia; alialsoweide@gmail.com (A.A.); helga1121@yandex.ru (O.K.); guliy_olga@mail.ru (O.G.)

**Keywords:** compact acoustical analyzer, resonator with lateral electric field, express analysis of antibiotics, bacterial cells, light phase-contrast microscopy, microbiological analysis

## Abstract

In this study, a compact acoustic analyzer for express analysis of antibiotics based on a piezoelectric resonator with a lateral electric field and combined with a computer was developed. The possibility of determining chloramphenicol in aqueous solutions in the concentration range of 0.5–15 μg/mL was shown. Bacterial cells that are sensitive to this antibiotic were used as a sensory element. The change in the electrical impedance modulus of the resonator upon addition of the antibiotic to the cell suspension served as an analytical signal. The analysis time did not exceed 4 min. The correlation of the experimental results of an acoustic sensor with the results obtained using the light phase-contrast microscopy and standard microbiological analysis was established. The compact biological analyzer demonstrated stability, reproducibility, and repeatability of results.

## 1. Introduction

Chloramphenicol (*Amphenicols* class) (CAP), widely known among the aromatic antibiotics, was isolated from the culture liquid of *Streptomyces venezuelae* in 1947, and in 1949 its chemical formula (C_11_H_12_Cl_2_N_2_O_5_) was established [1]. The CAP is the first antibiotic, which is commercially synthesized chemically, while most other antibiotics are biosynthesized. This has been largely facilitated by the relatively simple chemical structure of the antibiotic (Figure 1), which belongs to the group of para-nitrobenzene derivatives.

CAP is a broad-spectrum drug, because it is effective against most Gram-positive and Gram-negative microorganisms such as *Escherichia*, *Salmonella*, *Pasteurella*, *Staphylococcus*, *Streptococcus*, *Diplococcus*, *Proteus*, etc. The CAP acts on bacterial strains that are resistant to penicillins, streptomycin, and sulfonamides, and is active against gram-negative anaerobes. However, it is ineffective with respect to acid-fast bacteria, *Clostridia*, and *Pseudomonas aeruginosa* [2]. It is known that the resistance of microorganisms to the CAP develops relatively slowly. Because the CAP has serious side effects, it is used when other chemotherapeutic agents are ineffective. Due to this, the part of sales of *Amphenicols* relative to the total sales of antibiotics was increased from 1.9% in 2018 [3] to 2.3% in 2020 in EU countries [4]. It is important to note that this drug is actively used not only in medicine, but also in animal husbandry, due to its effective inhibition of bacterial growth. The widespread use of CAP, as well as the imperfection of the systems for its detection and wastewater treatment of the enterprises of various profiles, has led to the large-scale pollution of the natural objects and ecosystems by this compound and its derivatives. This significantly worsens the sanitary and hygienic state of the environment, especially water resources. According to the Food and Drug Administration, CAP is classified as pregnancy category C (classified as FDA pregnancy category C), requiring special care in its use [5]. In addition, CAP is highly toxic [2,6] and its use in some countries, including the United States, is limited by the infections for which the potential benefit of the antibiotic exceeds the risks of its use. Therefore, CAP is still used in developing countries due to its high efficiency and low cost [7,8]. The main danger of CAP is that after penetration in the environment, it does not decompose quickly, which leads to long-term environmental pollution [9,10]. The presence of CAP in the environment has a serious adverse effect on human health, since the drug quickly penetrates the body and affects the internal organs—primarily the liver [2]. Therefore, the content of CAP in environmental objects, especially water resources, is under close observation of environmentalists and is subject to mandatory monitoring.

An urgent problem of the environmental control of CAP content is the development of specific and rapid methods for its monitoring in aquatic environments and wastewater from industrial enterprises. Currently, analytical methods for the detection of CAP are used, such as high performance liquid chromatography, gas chromatography, and liquid chromatography-mass spectrometry [11,12,13,14,15,16], as well as indirect competitive chemiluminescent enzyme immunoassay [17]. These methods have a number of limitations due to the need of using expensive and high-tech equipment with low sensitivity, long time costs, and complex analytical procedures. Therefore, the development of new sensory methods for the rapid detection of CAP is a very relevant direction. For example, to determine CAP, there are the surface-enhanced Raman scattering sensors [18], electrochemical aptamer sensors [19,20,21], electrochemical sensors based on a graphene oxide [22], sensors based on a quartz crystal microbalance system with immobilized antibodies [23] amperometric [24], electrochemical [25], piezoelectric [26], optical [27], and photonic [28,29] sensors, as well as molecularly imprinted polymers sensing systems [30,31,32,33]. The general approaches for the analysis of the chloramphenicol are schematically presented in Figure 2.

An effective alternative to these methods of CAP environmental control in the aquatic environment is the use of biosensors based on microbial cells that are sensitive to CAP. The biosensors based on the whole cells have found application in the environmental monitoring and express analysis of aqueous media [34,35,36,37,38]. The change in the integrity of the bacterial cell membrane in response to the antibiotic contact action is accompanied not only by the membrane damage, but also by the redistribution of the cell surface charges and an increase in the conductivity of the measurement medium, which can be recorded by an acoustic sensor. The principle of the sensor is based on the registration of the bio specific reactions in a liquid suspension in contact with the surface of a piezoelectric wave-guide, through which a piezoactive acoustic wave propagates.

Previously, we have shown the possibility of analyzing antibiotics directly in a liquid using a sensor based on a resonator with a lateral exciting electric field [39]. Although the sensor itself is small (60 × 40 × 20 mm^3^), the precision LCR meter 4285A (Agilent, Santa Clara, CA, USA) was used to measure the sensor parameters. This meter has a high accuracy (up to 0.1%), but is very expensive and bulky. More promising are the compact analyzers that allow the measurements of biological objects in “field conditions” or small mobile laboratories. Therefore, a special device was developed, which has a somewhat lower accuracy, but is distinguished by the ease of manufacture, portability, and low cost.

The purpose of this paper was to study the possibility of using the developed compact acoustic analyzer for chloramphenicol analysis in the aquatic environment.

## 2. Materials and Methods

### 2.1. Bacteria and Culture Conditions

We used the bacterial cells *Escherichia coli* K-12 and *Escherichia coli* pBR-325, which were taken from the Collection of Rhizosphere Microorganisms of the Institute of Biochemistry and Physiology of Plants and Microorganisms—Subdivision of the Federal State Budgetary Research Institution Saratov Federal Scientific Centre of the Russian Academy of Sciences (IBPPM RAS) (http://collection.ibppm.ru, accessed on 25 February 2022).

An important task in the development of a sensor system is the selection of a bio selective agent (recognition element) providing the specific interaction with the target analyte, which will be recorded by the sensor. The microbial cells that are sensitive to the antibiotic being determined can be used as a sensor element of the analyzer. The biological activity of CAP is largely determined by the ability to inhibit the protein synthesis in microbial cells. Since CAP is a broad-spectrum antibiotic and is active against a large number of gram-negative rods, *E. coli* microbial cells with a concentration of 10^4^ cells/mL were used as the object of the study. In the preliminary microbiological experiments, it has been shown that these cells are sensitive to CAP.

For the cultivation of the bacteria, the liquid nutrient medium LB of the following composition (g/L) was used: NaCl (Company «LenReactive», St. Petersburg, Russia)—5.0; peptone (Becton, Dickinson & Co., Franklin Lakes, NJ, USA)—10.0; yeast extract (DIFCO, Franklin Lakes, NJ, USA)—5.0. The solid medium contained 1.5% and 3% agar-agar.

The cell suspension was prepared on the basis of distilled water with a conductivity of ~4 μS/cm and PH = 7.

### 2.2. Light Phase Contrast Microscopy

To control the effect of antibiotics on microbial cells, we used a laser dissector Leica LMD 7000 (Leica Microsystems, Wetzlar, Germany) with the TL-PH phase contrast method and magnification of 40. The research was carried out at the Simbioz Center for the Collective Use of Research Equipment in the Field of Physical-Chemical Biology and Nanobiotechnology (IBPPM RAS, Saratov, Russia). The cell preparation and sample scanning were performed as described in [40].

### 2.3. Antibiotics

In this work, we used the chloramphenicol of the company Sigma (St. Louis, MO, USA).

### 2.4. Determination of the Viability of the Bacteria

To calculate the number of colonies formed from the individual viable cells after the effects of the antibiotic, we used the standard method of seeding to the surface of the LB medium, as described in [41]. In the experiments done to assess the effect of the chloramphenicol, we used the antibiotic concentrations in the interval 0.5–15 μg/mL. The calculation data of the growing colonies during crossing cells without processing with an antibiotic was used as the control.

### 2.5. Preparation of Cells before Carrying Out the Measurements

Before analysis, the cells were washed three times with the culture medium by centrifugation at 2800× *g* for 5 min, and were then resuspended in a small amount of distilled water (electrical conductivity of 1.8 µS/cm). To eliminate the appearance of the conglomerates, the cell suspension was again centrifuged at 110× *g* for 1 min and the suspension remained in the supernatant that was used. Then, we brought the optical density of the prepared suspension *D*_670_ to 0.4–0.42 using a Specol photoelectron colorimeter (Carl Zeiss, Oberkochen, Germany).

### 2.6. Description of the Analyzer and Technique of Experiments

The acoustic sensory platform consists of two components: a sensitive biological element and a detection system that allows you to register the concentration or activity of various analytes presented in the sample. The detection system includes a resonator with a lateral electric field, based on a plate of lithium niobate of X-cut of 0.5 mm thick. Two aluminum electrodes with shear dimensions of 5 × 10 mm^2^ with a gap of 2 mm are deposited on the lower side of the plate. The liquid container, made of plexiglass, is glued to the top side of the plate with a sealant. For suppression of the parasitic oscillations, the part of the electrodes is covered by damping layers [42]. The resonator is connected to a measuring device based on the ARDUINO electronic designer. This device allows you to measure the dependence of the module of the electrical impedance of the resonator on the frequency and transmit data to the personal computer. The measuring device consists of a harmonic signal generator, a meter of RF voltage on a resonator with a lateral electric field and a meter of RF current flowing through this resonator. Thus, the signal from the generator was applied to a circuit consisting of the serially connected known resistance *R* and unknown impedance *Z* of the resonator, the value of which was necessary to determine. The numerical values of these parameters, as well as the frequency value of the generator *f*, were transmitted to a personal computer connected to the device. The special program included in the computer allowed to determine the module of the electrical impedance of the resonator *Z*, as the function of the frequency *f*. It should be noted that the device, including the lateral electric field excited resonator, measuring device, and computer program, are not commercial and they are home-made in the laboratory.

Figure 3a presents the exterior of the compact analyzer, which includes a sensor based on a resonator with a lateral electric field, a measuring device, and a personal computer. Figure 3b shows the sensor diagram of the resonator with a lateral electric field with a liquid container, which represents the plate of lithium niobate of X-cut with two electrodes and damping layers for the suppression of parasitic oscillations [42].

For analysis, a prepared suspension of the microbial cells with a concentration of 10^4^ cells/mL was introduced into a liquid container of the compact bioanalyzer and the frequency dependence of the sensor’s electrical impedance module was measured. Then the antibiotic was added, and the measurement was repeated. In all experiments, we used one concentration of the microbial cells (10^4^ cells/mL) and a different concentration of the antibiotic (from 0.5 to 15 μg/mL). The analytical signal was a change in the module of the electrical impedance of the sensor after adding the antibiotic in the suspension of the microbial cells. The choice of the pointed values of the concentration of the drug is due to the data on the minimum inhibitory concentration of the CAP with respect to *E. coli* cells [43]. The idea of the experiments was to assess the effect of the chloramphenicol on bacteria using a compact analyzer based on a piezoelectric resonator with a lateral electric field and compare the results obtained with the ones of the standard microbiological seeding and light phase contrast microscopy data.

All experiments were carried out at least five times. The relative error of the results of the measurements of the studied samples was ±2%. This means that when several identical experiments are carried out, the electrical impedance modulus have a spread of the values at any frequency within 2%. The measurements were carried out in a laboratory at a temperature of 25–27 °C, a pressure of ~750 mm Hg, and a humidity of ~40%.

## 3. Results and Discussion

When the chloramphenicol was added to a suspension of the *E. coli* K-12 microbial cells, a decrease in the electrical impedance modulus of the resonator was observed. The recording time of the analytical signal was ~4 min. Figure 4a–f show the frequency dependences of the electrical impedance modulus of the sensor with a suspension of *E. coli* K-12 cells before (curve 1) and after (curve 2) adding an antibiotic with the concentrations 0.5, 2.5, 5, 10, 12, and 15 µg/mL. As can be seen from the data presented in Figure 4a, the electrical impedance modulus changed already at the minimum used concentration of CAP in the sample (0.5 µg/mL).

Thus, after the addition of an antibiotic to a cell suspension, the modulus of the electrical impedance decreases, and the degree of the reduction rises by increasing the CAP concentration. This behavior indicates that the addition of an antibiotic increases the conductivity of the suspension [44].

The data obtained allowed to plot the dependence of the absolute value of the change in the electrical impedance modulus |Δ*Z*| from the CAP concentration. Figure 5a shows this dependence at a frequency of 6.65 MHz.

One can see that with an increase in the specific amount of the antibiotic introduced into the cell suspension, the change in the electrical impedance modulus increases and reaches saturation from an antibiotic concentration of 12 μg/mL.

We have made a linear interpolation of the dependence presented in Figure 5a. This linear relationship, shown in Figure 5b, can be represented by the equation
(1)|ΔZ|=0.289+0.106n
where |Δ*Z*| is the change in the electrical impedance module after adding an antibiotic (kOhm), and *n* is an antibiotic concentration (μg/mL). Hence, by measuring the change in the electrical impedance module |Δ*Z*|, one can define *n* as
(2)$$n=\frac{\left|\Delta Z\right|-0.289}{0.106},$$

Independent experiments were carried out to determine the concentration of an antibiotic using a calibration curve represented by Equations (1) and (2). Three samples of the aqueous solution of the chloramphenicol with different concentrations were made and five independent experiments were carried out with each sample. The results of the experiments are presented in Table 1. It can be seen that the quadratic error of one measurement of the impedance module change |Δ*Z*| for each experiment does not exceed 2%. As for the concentration *n* of chloramphenicol, determined from the graduation Equation (2), the quadratic errors of one measurement are 3.76, 2 and 2.1% for concentrations 4.53, 7.02, and 9.01 μg/mL. Thus, the error of one measurement of the antibiotic concentration does not exceed 4%, i.e., the reproducibility of the results lies within 4%.

To explain the results obtained, we consider the processes that occur with cells under the influence of an antibiotic. The antibiotic chloramphenicol is an inhibitor of the protein synthesis, the main action mechanism of which is associated with a violation of the protein synthesis at the stage of amino acid transfer from aminoacyl-tRNA to ribosomes [2]. The antimicrobial action of the antibiotic does not depend on the growth phase of the microorganism. With a sufficient concentration and duration of the exposure, the ribosomes destroy and the macromolecules leave the cell. This leads to an increase in the conductivity of the suspension, which is recorded by the analyzer. At low concentrations, the chloramphenicol significantly inhibits the protein synthesis in the sensitive bacteria. High concentrations of the antibiotic inhibit the processes of the respiration, formation, and accumulation of glutamic acid and phenylalanine, and block the synthesis of the nucleic acids [2]. Thus, the gradual increase in the value of the analytical signal by increasing the antibiotic concentration is explained by the gradual inhibition of the protein synthesis upon the release of the macromolecules.

Obviously, the results of determining the number of the viable cells after exposure to the drug can serve as a quantitative assessment of the antibiotic activity against the studied bacteria, obtained by a piezoelectric sensor. For this purpose, the CAP effect on bacteria was evaluated using a light phase contrast microscope (LMD). The choice of this microscopy may be explained by the fact that LMD allows distinguishing the relevant cells and provides the non-contact and contamination-free isolation of the individual cells. The high numerical aperture of the lens objectives and the short laser wavelength provide the images of the high resolution along the optical and shear directions. The possibility of using a laser dissector to analyze the antibiotics effect on the bacteria was demonstrated in [40]. Figure 6 shows that the number of the viable cells in the field of view decreases significantly by increasing the CAP concentration.

The comparison of the results obtained by an acoustic analyzer with the data of the microbiological seeding has shown that with an increase in the CAP concentration acting on bacteria, the value of the registered analytical signal increases, and the number of the viable bacteria decreases. Therefore, this strain can be recommended as a sensor element for the CAP analysis.

To determine the optimal analysis time, we studied the dynamics of the changes in the analytical signal of the sensor with the cell suspension depending on the time of the antibiotic exposure. The obtained frequency dependences of the electrical impedance modulus of the compact analyzer with the antibiotic concentration of 2 mg/mL for the exposure times of 4, 8, 12, 16, 20, and 24 min are shown in Figure 7.

These data allowed constructing the dependence of the absolute value of the change in the electrical impedance modulus *Z* on the time of the exposure of the cells to the chloramphenicol with a concentration of 2 mg/mL near the resonant frequency of 6.65 MHz (Figure 8). Figure 8 shows that with an increase in the time of the antibiotic exposure to bacteria, the change in the module of the electrical impedance of the sensor slightly decreases. After 4 and 24 min of the CAP exposure to bacteria, the changes in the module of the electrical impedance of the sensor are equal to 0.672 and 0.584 kOhm, respectively. Since the difference between these values is approximately 8% and the method being developed is focused on a quick response about the presence/absence of a detectable antibiotic, it can be recommended to carry out the analysis within 4 min.

It is known that in the course of the sensory analysis involving the microbial cells, there are a number of the nonspecific factors that can lead to a change in the recorded sensor signal, such as the nonspecific effect of the compounds due to the sorption of high or low molecular weight compounds on the cell surface. To confirm that the fixed changes in the analytical signal are associated with the sensitivity of the bacteria to the CAP, an experiment was carried out with the chloramphenicol-resistance cells, which contain the pBR-325 plasmid carrier resistant to the CAP. The conditions of the experiments were similar as for a sensitive strain. Figure 9 presents the frequency dependences of the electrical impedance module of the sensor after the CAP addition of 2.5 μg/mL (a) and 10 μg/mL (b) in the suspension of the chloramphenicol-resistance cells. From the presented dependences, it can be seen that in both cases the electrical impedance module of the sensor does not practically change. Consequently, the results for the sensitive and resistant strains of *E. coli* are significantly different. This proves the selectivity of the developed compact sensory system to a certain antibiotic and the absence of the CAP non-specific influence on the cells and analytical signal value.

To eliminate the effect of the mass loading on the resonator associated with the introduction of an antibiotic to the container, an additional experiment was carried out in which the CAP with the concentration of 10 μg/mL was added to the container with distilled water without bacterial cells. These results are presented in Figure 10. One can see that the registered sensor parameters in this case practically coincide. Consequently, the change in the mass of water after the addition of an antibiotic does not lead to a change in the module of the electrical impedance of the resonator.

Thus, we have shown the possibility of a compact acoustic analyzer for CAP detection in aqueous media with the lower limit of a detection of 0.5 μg/mL, using the microbial cells sensitive to the chloramphenicol as a sensory element. The use of an acoustic method for analyzing the effects of CAP on bacterial cells creates new opportunities for solving the various biotechnological tasks, including not only the determination of the antibiotic, but also the estimation of its antibacterial activity.

## 4. Conclusions

Chloramphenicol is a broad-spectrum antibiotic that has been used in the past in veterinary medicine to treat all major farm animals and is currently used to treat humans and domestic animals. However, there have always been concerns about the genotoxicity of the chloramphenicol and its metabolites, its embryo- and fetotoxicity, its carcinogenic potential in humans, and the lack of a dose-response relationship for the aplastic anemia induced by the chloramphenicol treatment in humans [45]. The growing urbanization, the growth of production and consumption of the chloramphenicol despite existing prohibitions, are the main factors responsible for the appearance of this drug in the environment and, especially, in the water resources. The express assessment of the pollution of the environmental objects is currently a necessary component of the environmental research, so sensor technologies are being actively developed. This paper presents a microbial sensor system based on a compact acoustic analyzer for CAP determination with a lower limit detection of 0.5 μg/mL. This is significantly below the threshold (0.3 µg/kg) of the minimum content of CAP in the water and food by the Rapid Alert System for Food and Feed (RASFF).

Previously, when studying the possibility of detecting kanamycin in a liquid, a sensor based on a resonator with a lateral electric field was connected to the precision LCR meter 4285a (Agilent, Santa Clara, CA, USA) [39]. This meter has a high accuracy (up to 0.1%), but is very expensive and bulky and allows measuring biological objects only in laboratory conditions. Therefore, a special compact analyzer has been developed, which has a slightly lower accuracy, but is characterized by its simplicity of manufacture, portability, and low cost. The described compact acoustic analyzer, including the resonator with a lateral electric field and microbial cells, allows one to analyze CAP in aqueous solutions in real time and in situ on a large number of samples in “field conditions” and in mobile laboratories.

## Figures and Tables

**Figure 1 sensors-22-02937-f001:**
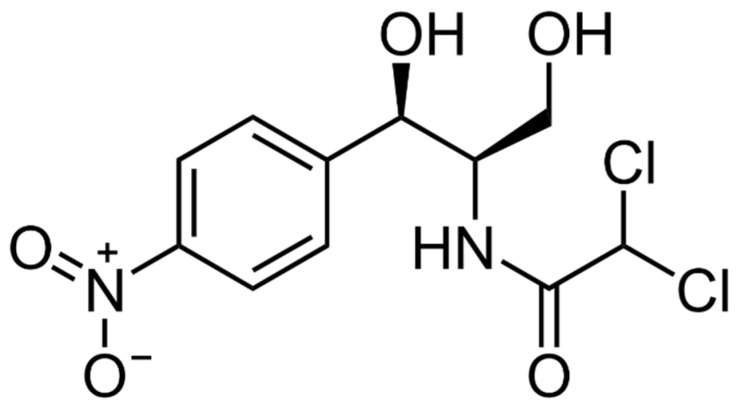
The structural formula of chloramphenicol.

**Figure 2 sensors-22-02937-f002:**
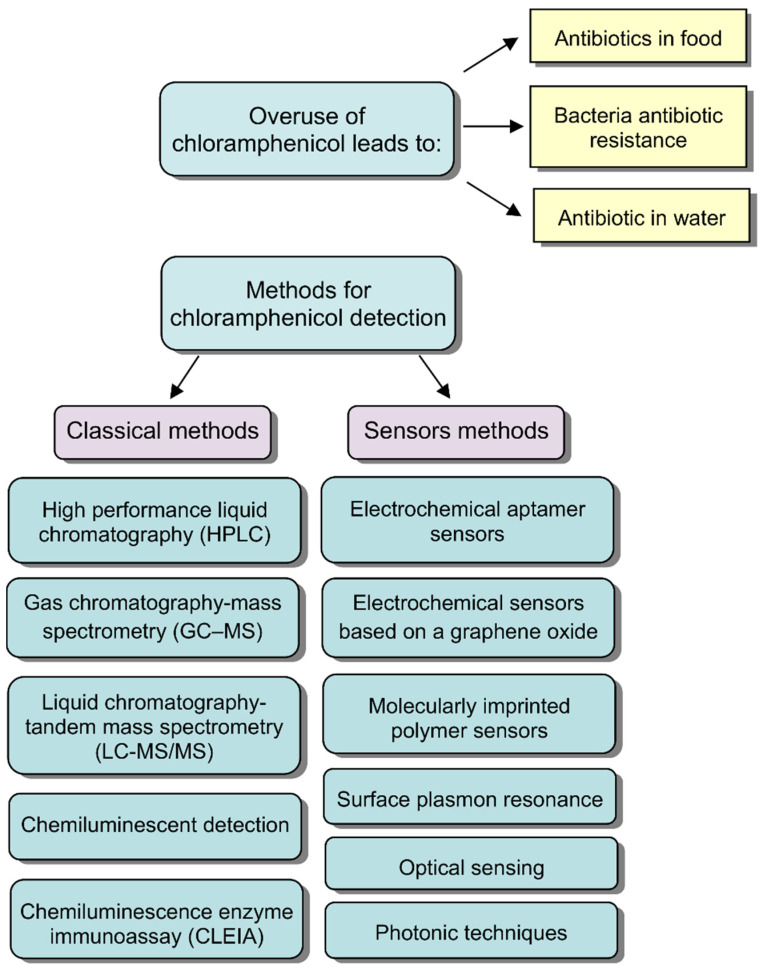
The general methods for the chloramphenicol analysis.

**Figure 3 sensors-22-02937-f003:**
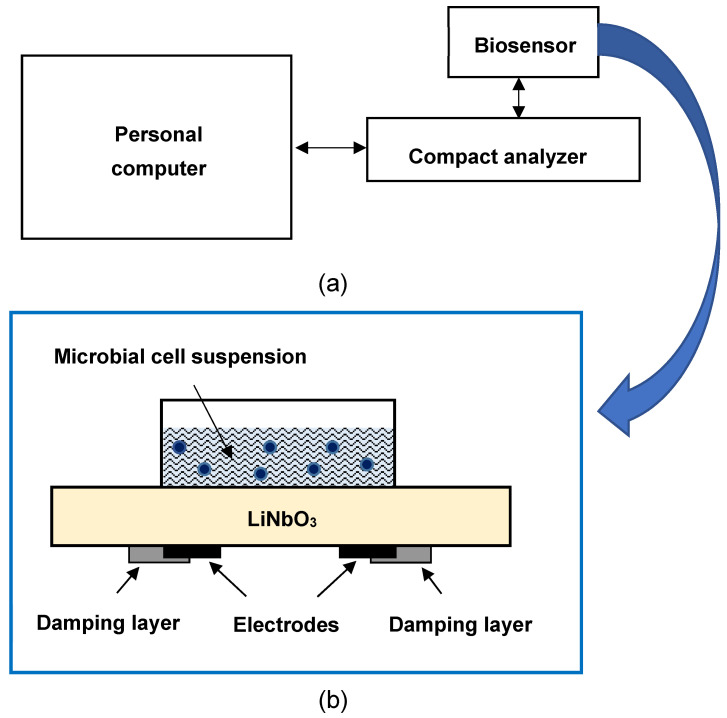
(**a**) The scheme of compact acoustic analyzer for antibiotic analysis in aqueous solutions; (**b**) the sensor based on a resonator with a lateral electric field with a liquid container.

**Figure 4 sensors-22-02937-f004:**
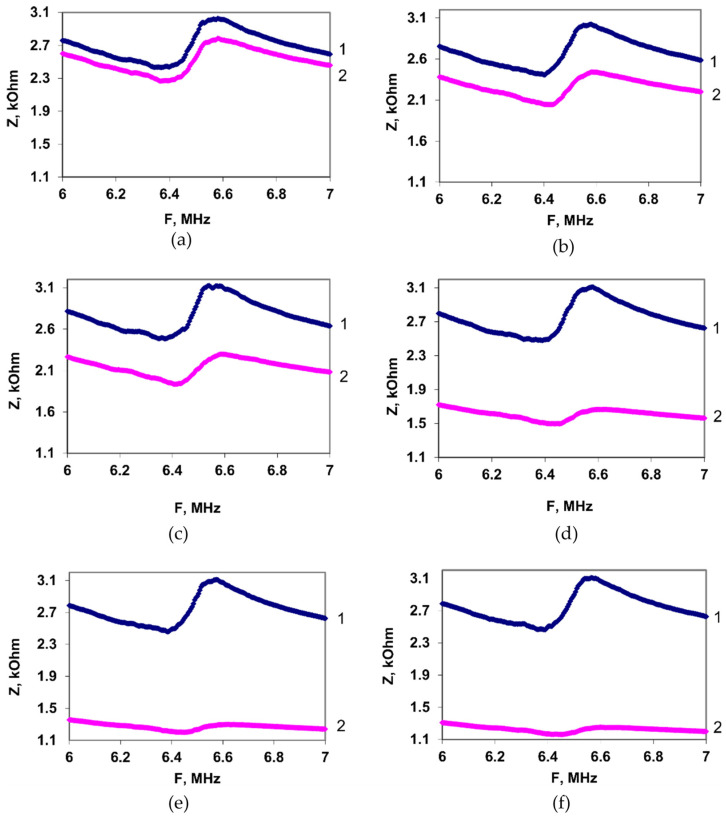
The dependences of the electrical impedance modulus *Z* on the frequency for a suspension of *E. coli* K-12 cells before (curve 1) and after (curve 2) the addition of the chloramphenicol with the concentration (µg/mL): (**a**)—0.5; (**b**)—2.5; (**c**)—5; (**d**)—10; (**e**)—12; (**f**)—15.

**Figure 5 sensors-22-02937-f005:**
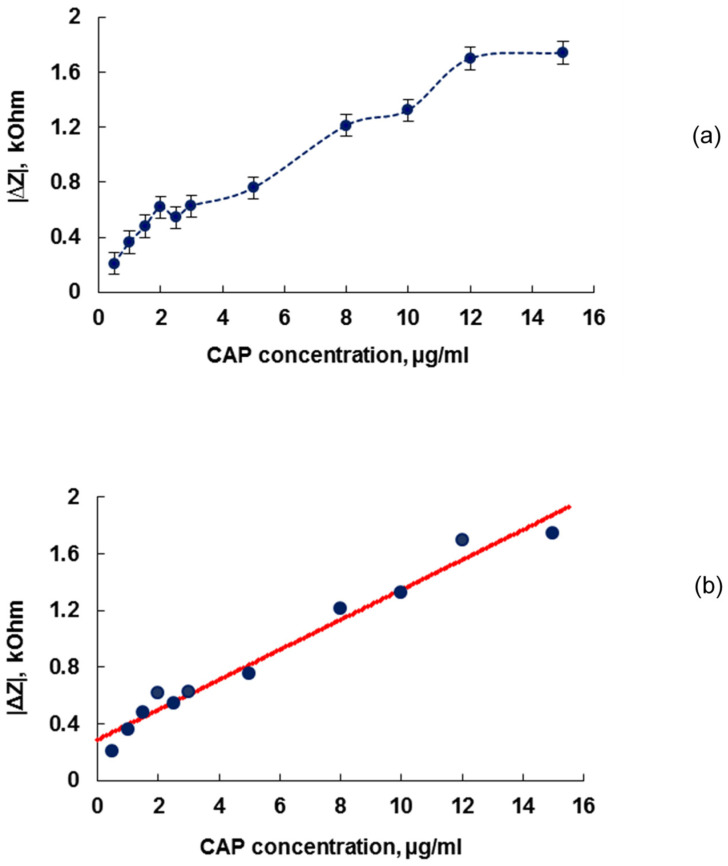
The dependence of the absolute value of the change in the electrical impedance modulus |Δ*Z*| on the CAP concentration added to *E. coli* K-12 cells at a frequency of 6.65 MHz without (**a**) and with (**b**) the calibration curve.

**Figure 6 sensors-22-02937-f006:**
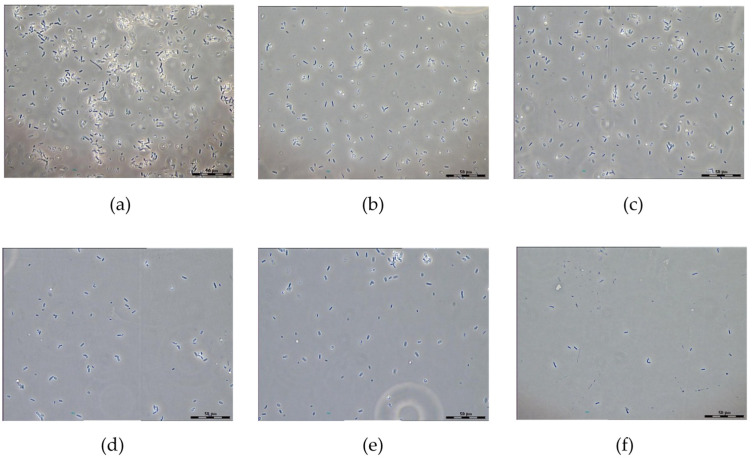
The laser dissector images of *E. coli* K-12 before (**a**) and after the CAP treatment with the different concentrations (μg/mL): (**b**)—0.5; (**c**)—2.5; (**d**)—5; (**e**)—10; (**f**)—15. Scale bar is 50 µm.

**Figure 7 sensors-22-02937-f007:**
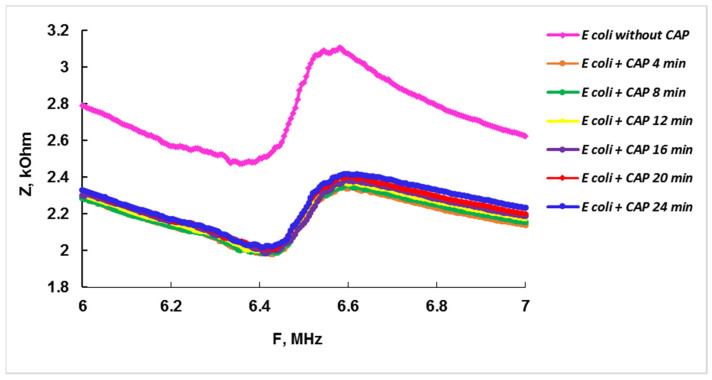
The frequency dependences of the electrical impedance modulus of a compact analyzer with a suspension of *E. coli* K-12 cells before (pink curve) and after the CAP exposure with the concentration of 2 mg/mL for 4, 8, 12, 16, 20, and 24 min.

**Figure 8 sensors-22-02937-f008:**
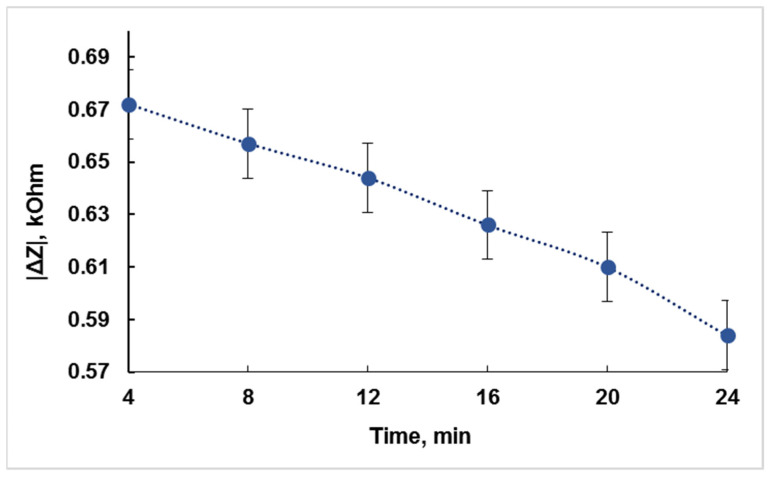
The dependence of the change in the absolute value of the electrical impedance modulus |Δ*Z*| on the time of the CAP exposure to *E. coli* K-12 cells at a frequency of 6.65 MHz.

**Figure 9 sensors-22-02937-f009:**
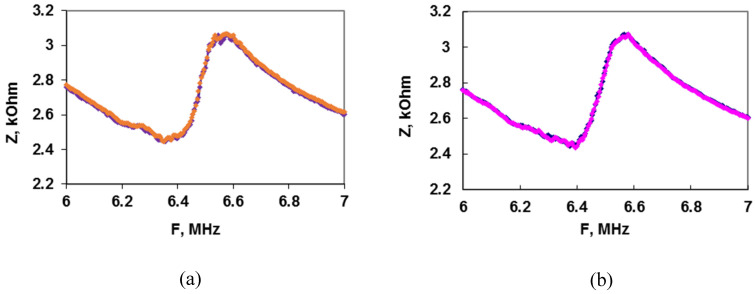
The frequency dependences of the module of the electrical impedance *Z* for the *E. coli* pBR-325 cells before (blue curves) and after (orange and pink curves) the CAP treatment with the concentration (μg/mL): 2.5 (**a**) and 10 (**b**).

**Figure 10 sensors-22-02937-f010:**
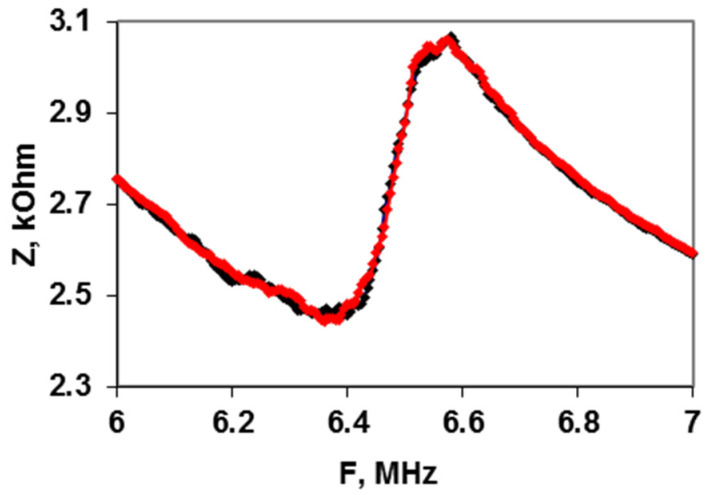
The frequency dependences of the electrical impedance module *Z* for distilled water before (black line) and after (red line) adding CAP of 10 μg/mL.

**Table 1 sensors-22-02937-t001:** The results of the independent experiments for the measurement of chloramphenicol concentration using a calibration curve in Figure 5b. We have introduced the following designations: |Δ*Z*| is the module of the impedance of the resonator; |Δ*Z*_av_| is the averaged module of the impedance of the resonator; n is the CAP concentration; n_av_ is the averaged value of CAP concentration; Δn_av_ is the deviation from n_av_; Sq is the squared error of one measurement; Res1 is the result of one measurement; Res5 is the result of five measurements.

|Δ*Z*|, kHz	|Δ*Z*_av_|, kHz	n, µg/mL	n_av_, µg/mL	Δn_av_, µg/mL	Sq, µg/mL	Res1	Res5
0.764	0.764 ± 1.6%	4.5	4.534	0.034	0.17	4.534 ± 0.17 4.534 ± 3.76%	4.534 ± 0.08 4.52 ± 1.7%
0.779		4.79		−0.256			
0.749		4.35		0.184			
0.771		4.6		−0.066			
0.756		4.43		0.104			
1.028	1.022 ± 1.75%	7	7.02	0.02	0.135	7.02 ± 0.135 7.02 ± 2%	7.02 ± 0.06 7.02 ± 0.91%
1.05		7.2		−0.18			
1.009		6.85		0.17			
1.039		7.1		−0.08			
1.019		6.95		0.07			
1.241	1.24 ± 1.5%	9.05	9.01	−0.04	0.192	9.01 ± 0.192 9.01 ± 2.1%	9.01 ± 0.09 9.01 ± 0.95%
1.264		9.25		−0.24			
1.215		8.75		0.26			
1.252		9.1		−0.09			
1.227		8.9		0.11			

## Data Availability

Not applicable.

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
