# Peer review of "Microbial Acoustical Analyzer for Antibiotic Indication"

_sensors, 2022, doi:10.3390/s22082937_

Round 1

Reviewer 1 Report

The paper presents a microbial sensor based on a compact acoustic analyzer for the CAP determination with a lower limit detection of 0.5 μg/ml which is  significantly below the threshold (0.3 μg/kg) of the minimum content of the CAP in the water and food by the Rapid Alert System for Food  and Feed (RASFF).

This is a very important result!

The paper is well written and clear. The presentation is very good and the English is fine. I am only surprised that figures are included in the introduction that is rare to see, but this is up to the MDPI rules.

As to me, the paper can be published as it is.

Author Response

Thank you for appreciating our work

Reviewer 2 Report

The Authors have presented a very interesting microbial sensor system based on a compact acoustic analyzer for the Chloramphenicol determination. The proposed application is very remarkable in particular thanks to the small size of the entire measuring system, which allows very simple operations.

The paper is well-organized and the argumentations are well supported by experimental procedures and data.

In order to make the work more complete, I think it is appropriate to add some details about the used acoustic device. Figure 3 is certainly clarifying but the details on the characteristics of the sensor would be interesting. Please specified if the device is commercial, home-made or others.

Author Response

Thanks for your comment.

We expanded the description of the resonator on page 4: “The detection system includes a resonator with a lateral electric field, based on a plate of lithium niobate of X – cut of 0.5 mm thick. Two aluminum electrodes with shear dimensions of 5×10 mm2 with a gap of 2 mm are deposited on the lower side of the plate. The liquid container, made of plexiglass, is glued to the top side of the plate with a sealant. For the suppression of the parasitic oscillations the part of electrodes was covered by damping layer [42]. The resonator is connected to a measuring device based on the ARDUINO electronic designer.”

We have added also the following sentence on the page 5:“It should be noted that the device including the lateral electric field excited resonator, measuring device and computer program are not commercial and they are home-made in laboratory.”

Reviewer 3 Report

The purpose of this study was to develop a microbial acoustical analyzer for antibiotic indication. This is an incomplete study and could not be recognized as a research paper in this stage.

  1. The environment conditions should be noted in this study, what is the temperature, pH and other ion concentrations.
  2. By the information of Figure 5, the calibration equation could be established.
  3. The independent test must be done. That is, prepare the other concentrations, performed the test and find the measurement concentrations with this calibration equations.
  4. Define the stability, reproducibility and repeatability, and supply the actual data to prove these performance.
  5. Perform the field test to confirm the statement in the abstract “The possibility of using in "the field conditions" and in the mobile laboratories.”

Author Response

1.The environment conditions should be noted in this study, what is the temperature, pH and other ion concentrations.

Thank you for this comment.

We have included on the page 4 the following sentence: “The cell suspension was prepared on the basis of distilled water with a conductivity of ~4 μS/cm and ph =7.”

We have also included on the page 5 the following sentence: “The measurements were carried out in a laboratory at a temperature of 25 – 27°C, a pressure of ~750 mm Hg, and a humidity of ~40%.”

2.By the information of Figure 5, the calibration equation could be established.

Thank you for this comment. We have introduced on the page 7 the following text:

We have made a linear interpolation of the dependence presented in Fig.5a.This linear relationship, shown in Fig.5b can be represented by the equation

              (1)

where |ΔZ| is the change in the electrical impedance module after adding an antibiotic (kOhm), n is an antibiotic concentration (μg/ml).Hence by measuring the change in the electrical impedance module |ΔZ| one can define n as

                 (2)

The equations are presented in attachment file.

3.The independent test must be done. That is, prepare the other concentrations, performed the test and find the measurement concentrations with this calibration equations.

4.Define the stability, reproducibility and repeatability, and supply the actual data to prove these performance

Thank you for this comment. We have introduced on the pages 7-8 the following text:

“Independent experiments were carried out to determine the concentration of an antibiotic using a calibration curve represented by equations (1 – 2). Three samples of the aqueous solution of chloramphenicol with different concentrations were made and 5 independent experiments were carried out with each sample. The results of the experiments are presented in Table 1. It can be seen that the quadratic error of one measurement of the impedance module change |ΔZ| for each experiment does not exceed 2%. As for the concentration n of chloramphenicol, determined from the graduation equation (2), the quadratic errors of one measurement are 3.76, 2 and 2.1% for concentrations 4.53, 7.02 and 9.01 μg/ml. Thus, the error of one measurement of the antibiotic concentration does not exceed 4%, i.e. the reproducibility of results lies within 4%.”

We have also included the Table 1 with the results of these experiments (in revised manuscript).

5.Perform the field test to confirm the statement in the abstract “The possibility of using in "the field conditions" and in the mobile laboratories.”

Thank you for this comment.

We had in mind the possibility of measuring in various industrial and residential rooms at a normal conditions at a temperature of 20-30 C, and not literally in fields and forests both in winter and in summer. We have removed this phrase from the abstract.

Reviewer 4 Report

A seemingly interesting paper, although I fund very hard to understand what the authors are measuring?

Impedance in a 3-electrode cell configuration? Or resistance of the piezoelectric material? If it is resistance of a solid, why impedance? The resistance of this material id frequency dependent? Why not simply using a digital multimeter? 

Why the choice of Mhz frequencies?

Author Response

Thank you for your comment. The answers are presented below and are not included in the text of the paper.

In this case, the sensor is a piezoelectric resonator, consisting of a piezoelectric plate and two electrodes located on the bottom of the plate.An alternating electrical voltage applied to the electrodes causes mechanical oscillations of the plate due to the piezoelectric effect, which have the greatest intensity at the resonant frequency.In turn, the resonant frequency depends on the geometric dimensions of the resonator and on the mechanical, piezoelectric and dielectric properties of the plate.Mechanical oscillations are accompanied by an alternating electric field that penetrates into the medium contacting with the free side of the plate. Consequently, a change in the electrical conductivity of this medium will change the characteristics of the resonator, which is a combination of a plate with electrodes and a contacting medium (cell suspension). Thus, the electrical impedance of the resonator is equal to the ratio of the amplitude of the voltage applied to the electrodes to the amplitude of the current flowing through the resonator. In general, impedance is complex, i.e. it consists of real and imaginary parts. In this case, we use the impedance module, i.e. square root from the sum of squares of real and imaginary parts.

Why not simply using a digital multimeter? Why the choice of Mhz frequencies?

Conductometric methods are widely used in biology for the analysis of cell suspensions [Stewart, G.N. (1899). "The changes produced by the growth of bacteria in the molecular concentration and electrical conductivity of culture media". Journal of Experimental Medicine. 4: 235–243. doi:10.1084/jem.4.2.235; Parsons, L.B.; Sturges, W.S. (1926). "The possibility of the conductivity method as applied to studies of bacterial metabolism". Journal of Bacteriology. 11: 177–188: Chiara Canali, et al, Conductometric analysis in bio-applications: A universal impedance spectroscopy - based approach using modified electrodes, Sensors and Actuators, 2015, vol. B212, pp.544-550]. The cheapest conductivity meter uses a constant electric current flowing through the suspension. What are the disadvantages with this approach. First, there is a polarization of the electrodes, which distorts the measurement results. In this case, it is necessary to take into account the polarization of the electrodes, which is quite a complicated problem. It is possible to increase the area of the electrodes, which will lead to the need to use the more amount of the suspension. In addition, at a constant current, the most part of the current flows through the intercellular spaces without reacting to a change in the conductivity of the cytoplasm of the cell under the action of the added reagent. Finally, electrolysis is observed in this case, which also leads to great difficulties in interpreting the data.

Therefore, in biology, the alternating currents at the frequencies up to several megahertz are used. In this case, the total conductivity is determined both by intercellular spaces and by the cytoplasm of the cell, which can also vary under the action of the added reagent. Obviously, such a conductometer consists of an RF voltage generator and a corresponding bridge circuit, i.e. is not so simple and cheap.

In our case, the frequency of the electric field applied to the suspension lies in the range 6-7 MHz. This means that the sensor catches the change in conductivity both in the intercellular space and in the cytoplasm inside the cells. In addition, the use of the resonance effects leads to an increasing the sensitivity of the analysis of the changes in cell suspension. Finally, biological reactions can lead to a change in both the permittivity and viscosity [Takashi Kogai, et al, Rayleigh SAW assisted SH-SAW immunosensor on X-cut 148-Y LiTaO3, IEEE Trans. On Ultrason., Ferroel., and Freq. Control. 2017, vol.64, No 9, pp. 1375-1381] and the conductivity meters will not notice changes in these parameters. The sensor described in the article is sensitive to these changes and therefore is more promising. Finally, our sensor allows us to quickly remove the waste microorganisms from liquid container, which does not containmetal electrodes. For a conductivity meter with porous electrodes, cleaning is a rather complex problem.

Round 2

Reviewer 3 Report

The revised manuscript have been improved significantly.

Reviewer 4 Report

I believe the manuscript has been significantly improved and should be published in its present form after some minor language editing.